# Association of the Serotonin and Kynurenine Pathways as Possible Therapeutic Targets to Modulate Pain in Patients with Fibromyalgia

**DOI:** 10.3390/ph17091205

**Published:** 2024-09-12

**Authors:** Alfonso Alfaro-Rodríguez, Samuel Reyes-Long, Ernesto Roldan-Valadez, Maykel González-Torres, Herlinda Bonilla-Jaime, Cindy Bandala, Alberto Avila-Luna, Antonio Bueno-Nava, Elizabeth Cabrera-Ruiz, Pedro Sanchez-Aparicio, Angélica González Maciel, Ana Lilia Dotor-Llerena, José Luis Cortes-Altamirano

**Affiliations:** 1Division of Basic Neurosciences, Instituto Nacional de Rehabilitación, “Luis Guillermo Ibarra Ibarra”, Mexico City 14389, Mexicoernest.roldan@usa.net (E.R.-V.);; 2Department of Radiology, I.M. Sechenov First Moscow State Medical University (Sechenov University), 119992 Moscow, Russia; 3Conahcyt & Biotechnology Laboratory, Instituto Nacional de Rehabilitación, “Luis Guillermo Ibarra Ibarra”, Mexico City 03940, Mexico; mikegcu@gmail.com; 4Department of Reproductive Biology, Universidad Autónoma Metropolitana Iztapalapa, Mexico City 09340, Mexico; 5Escuela Superior de Medicina, Instituto Politécnico Nacional, Mexico City 11340, Mexico; 6Pharmacology Department, Facultad de Medicina Veterinaria, Universidad Autónoma del Estado de México, Toluca 50090, Mexico; 7Laboratory of Cell and Tissue Morphology, Instituto Nacional de Pediatría, Mexico City 04530, Mexico; 8Division of Clinic Neurosciences, Instituto Nacional de Rehabilitación, “Luis Guillermo Ibarra Ibarra”, Mexico City 14389, Mexico; 9Department of Chiropractic, Universidad Estatal del Valle de Ecatepec, Ecatepec de Morelos 55210, Mexico

**Keywords:** tryptophan, 5-HT1A, 5-HT2, 5-HT3, 5-HTT, quinolinic acid, kynurenine acid

## Abstract

Fibromyalgia (FM) is a disorder characterized by widespread chronic pain, significant depression, and various neural abnormalities. Recent research suggests a reciprocal exacerbation mechanism between chronic pain and depression. In patients with FM, dysregulation of tryptophan (Trp) metabolism has been identified. Trp, an essential amino acid, serves as a precursor to serotonin (5-HT), a neuromodulator that influences mood, appetite, sleep, and pain perception through the receptors 5-HT1, 5-HT2, and 5-HT3. Additionally, Trp is involved in the kynurenine pathway, a critical route in the immune response, inflammation, and production of neuroactive substances and nicotinamide adenine dinucleotide (NAD+). The activation of this pathway by pro-inflammatory cytokines, such as tumor necrosis factor α (TNF-α) and interferon gamma (IFN-γ), leads to the production of kynurenic acid (KYNA), which has neuroprotective properties, and quinolinic acid (QA), which is neurotoxic. These findings underscore the crucial balance between Trp metabolism, 5-HT, and kynurenine, where an imbalance can contribute to the dual burden of pain and depression in patients with FM. This review proposes a novel therapeutic approach for FM pain management, focusing on inhibiting QA synthesis while co-administering selective serotonin reuptake inhibitors to potentially increase KYNA levels, thus dampening pain perception and improving patient outcomes.

## 1. Introduction

Fibromyalgia (FM) is characterized by widespread and persistent pain, alongside significant fatigue and sleep disturbances. It is also closely related to a relatively high prevalence of several comorbid conditions, including irritable bowel syndrome, major depressive disorder, nervous bulimia, cataplexy, anxiety, obsessive–compulsive disorder, panic disorder, post-traumatic stress disorder, premenstrual dysphoric disorder, social phobias, migraines, and joint pain [1,2]. The global prevalence of fibromyalgia is approximately 1.78%, although it varies by region, with rates of 3.3% in North America and 2.9% in Europe. Women are disproportionately affected, with a ratio of 3:1 compared to men. Among women, prevalence estimates range from 0.2% to 6%, while urban populations exhibit rates between 2.4% and 6.8%. In certain populations, prevalence can range from 0.6% to 15%, depending on the diagnostic criteria applied. Typically, fibromyalgia patients are middle-aged women (30–60 years old) with a basic level of education [3]. Given the central role of chronic pain in FM, its global prevalence is noteworthy; in the United States alone, an estimated 50 million people live with chronic pain, ranging from intermittent occurrences to debilitating conditions that prevent employment [4]. The economic impact is substantial, with medical and home care expenses reaching approximately USD 500 million annually [5]. The primary treatment for fibromyalgia to inhibit pain involves a combination of pharmacological approaches, physical therapies, psychological support, and self-management strategies. This multimodal treatment plan is designed to address the diverse symptoms experienced by patients with fibromyalgia, ultimately improving their quality of life. These observations highlight the urgent need for improved diagnostic and therapeutic approaches for FM and related disorders. A systematic literature search was conducted across databases such as PubMed, Scopus, and Web of Science. The search included studies published between 1980 and 2023, using keywords like ‘fibromyalgia’, ‘pain mechanisms’, ‘serotonin’, and ‘tryptophan’. The inclusion criteria were limited to peer-reviewed articles, clinical trials, and meta-analyses, excluding case studies and editorials. Articles were screened based on their relevance to the pathophysiological mechanisms and treatment strategies for fibromyalgia, as discussed in this manuscript.

## 2. Physiopathology of Fibromyalgia

The pathophysiology of fibromyalgia (FM) is multifaceted, encompassing abnormalities in the nervous and neuroendocrine systems, autonomous system dysfunctions, genetic predispositions, psychosocial factors, and environmental stressors [6,7]. FM is closely associated with dysfunction in monoamine pathways, particularly concerning serotonin. Alterations in neurotransmitter systems, such as serotonin (5-HT) and norepinephrine (NE), are implicated in mood and pain disorders. Genetic polymorphisms, such as those in the serotonin transporter (5-HTTLPR), affect serotonin signaling, increasing susceptibility to conditions like fibromyalgia and depression. Neuroinflammation and dysregulation of the hypothalamic–pituitary–adrenal (HPA) axis also contribute to exaggerated pain responses and mood disorders, linking inflammation to monoamine dysfunction. Additionally, functional imaging studies reveal that certain serotonin-related genetic variants are associated with increased activation in brain regions responsible for pain perception. These mechanisms highlight the importance of the balance between neurotransmitters, such as serotonin, norepinephrine, and dopamine, and their impact on fibromyalgia [8]. Patients with FM exhibit heightened sensitivity to various stimuli, including temperature changes and mechanical and ischemic pressures, resulting in pain responses to stimuli that would typically be nonpainful for healthy individuals [9,10,11]. The primary source of pain in FM remains elusive, but emerging evidence points to a central nervous system (CNS)-mediated increase in sensory processing, akin to that observed in neuropathic pain conditions [12].

A growing body of research underscores the complex, bidirectional relationship between pain and depression, highlighting that shared pathophysiological pathways and molecular mechanisms are crucial to both conditions [13,14,15]. Notably, the prefrontal cortex, insular cortex, anterior cingulate cortex, amygdala, hippocampus, and thalamus have been implicated in the modulation of mood and pain perception. Postmortem analyses have revealed reductions in neuronal bodies within the prefrontal cortex, hippocampus, and thalamus, indicative of diminished synaptic activity [16]. Genetic research supports these findings, showing a decrease in the expression of genes related to excitatory synapses, particularly in these brain regions [17,18]. These discoveries suggest a potential link between chronic pain and depression progression through neuroplasticity alterations, underscoring the need for integrated therapeutic strategies that address the intertwined nature of these symptoms in FM.

## 3. Tryptophan and Fibromyalgia

Recent studies, including Groven et al. (2021) [19], have suggested that altered tryptophan (Trp) metabolism could be crucial in the pathology of fibromyalgia (FM). Trp, an indispensable amino acid, is the precursor of serotonin (5-hydroxytryptamine, 5-HT), a neurotransmitter integral to mood, appetite, and sleep regulation, and is increasingly recognized for its role in nociception [20]. Additionally, Trp is involved in the kynurenine pathway, which influences immune function, inflammation, and the biosynthesis of neuroactive compounds [21].

FM is characterized by reduced 5-HT levels and a notable decrease in the 5-HT transporter (5-HTT) within the cerebrospinal fluid of affected individuals [1,22,23,24,25]. Currently, metabolites of the kynurenine pathway, including quinolinic acid (QA) and kynurenic acid (KYNA), influence central nervous system functionality, acting as either neurotoxic or neuroprotective agents, particularly in modulating glutamatergic neurotransmission [19]. Therefore, disruptions to this metabolic pathway may manifest as pain in patients with FM.

Investigating the metabolic routes of Trp in patients with FM could elucidate its potential role in hyperalgesia, fatigue, depression, and other symptoms characteristic of this condition. The concept of a ‘cytokine storm’ triggered by tissue damage, leading to inflammation that adversely affects the brain, underscores the interconnectedness of these processes [26,27]. Such mechanisms, which are far from being isolated, play crucial roles in behavior development and can contribute to depression [28,29]. Moreover, chronic inflammation is associated with changes in neuronal structures and the metabolism of neurotransmitters related to pain, mood disorders, and cognition [29,30]. Therefore, Trp metabolism represents a critical intersection between the 5-HT and kynurenine pathways, which may influence the co-occurrence of pain and depression in patients with FM.

## 4. Role of Serotonin Receptors in Pain in Patients with Fibromyalgia

This section explores the potential link between serotonin (5-HT) and fibromyalgia (FM), highlighting the central role of serotonin as a monoamine neurotransmitter in the CNS that is synthesized from tryptophan (Trp). Serotonin acts as a neuromodulator in various physiological processes, including sleep, appetite, thermoregulation, hormone secretion, sexual behavior, and pain management [20,31,32,33]. The biosynthesis of 5-HT begins with the hydrolysis of Trp to 5-hydroxytryptophan (5-HTP) by tryptophan hydroxylase, followed by the conversion of 5-HTP to 5-HT through the action of aromatic L-amino acid decarboxylase. Trp is transported across the blood–brain barrier by a specific transporter in the raphe nuclei of the midbrain, which is projected throughout the brain [34].

The interplay between 5-HT activity and pain perception in patients with FM is intricate, and various studies indicate that disruptions to serotonin metabolism and transmission significantly impact nociception. The descending pain pathway is a complex neural system crucial for pain modulation, as it transmits signals from the brain to the spinal cord and influences pain perception. This pathway involves nociceptive transmission, where painful stimuli are conveyed from peripheral receptors to the brain for conscious perception. A key component of this modulation is serotonin, a neurotransmitter that inhibits pain signals in the spinal cord by reducing the release of substance P and regulating nociceptive responses. In conditions such as fibromyalgia, lower levels of serotonin are associated with increased pain sensitivity, underscoring the role of serotonin in pain modulation [35]. Altered nociception, a prevalent feature of FM, suggests a pivotal role for 5-HT in this pathology.

Research involving mirtazapine has demonstrated its potential in reversing chronic pain and improving the analgesic effects of morphine in FM-like mouse models. Mirtazapine works by increasing 5-HT release, activating the 5-HT1 receptor, and inhibiting the postsynaptic 5-HT2 and 5-HT3 receptors, as well as the presynaptic adrenergic α2 receptors. This action may reduce pain and enhance hypoalgesia in FM. While other medications, such as pregabalin, have been explored, mirtazapine presents a novel strategy that could offer additional benefits. Nonetheless, more studies are needed to comprehensively understand its effectiveness and potential side effects in humans [36].

### 4.1. 5-HT1 Receptor

Studies have shown that regulation of serotonin (5-HT) release in synapses is mediated by presynaptic autoreceptors, specifically 5-HT1A, and the serotonin transporter (5-HTT), leading to a reduction in serotonergic neuronal activity [37,38]. These presynaptic 5-HT1A metabotropic receptors, particularly within the corticolimbic system, play crucial roles in mediating antidepressive and antinociceptive effects through the reuptake of 5-HT, thus controlling its availability and the duration of its effects in the synaptic cleft [39,40,41,42]. In the context of fibromyalgia (FM), a decrease in 5-HT levels within cerebrospinal fluid, accompanied by an increase in 5-HTT activity, has been noted. This observation suggests a potential dysregulation of the serotoninergic system in FM, indicating a possible target for therapeutic intervention [43,44].

### 4.2. 5-HT2A Receptor

Research has increasingly indicated a link between fibromyalgia (FM) and the 5-HT2A metabotropic receptors, a critical element of pain perception and the central modulation of nociceptive pathways [45]. Genetic variations in the 5-HT2A receptor have been associated with FM, with certain genotypes correlated with heightened pain sensitivity [46]. Furthermore, decreased levels of 5-HIAA, a metabolite of 5-HT, in the serum have been linked to improved self-reported pain perception among FM patients, suggesting potential dysregulation of 5-HT [47].

The interconnection between FM and the 5-HT2A receptor in pain is intricate and is shaped by genetic variation and the deregulation of the serotonin pathway [45]. Recent findings have demonstrated that the modulation of 5-HT receptors, including 5-HT2A, may be effective in managing chronic pain in FM sufferers [36]. Cyclobenzaprine, an antagonist of the 5-HT2A receptor, has been shown to be beneficial for most sleep disorders and has shown a modest improvement in FM-related pain. This evidence supports the significant role that these receptors may play in FM [48,49,50]. Furthermore, studies suggest that downregulation of the 5-HT2A and 5-HT2C receptors could contribute to the onset of chronic hyperalgesia in FM [45,51]. These findings underscore the potential of targeting these receptors for the pharmacological treatment of pain in patients with FM, particularly those with comorbid conditions.

### 4.3. 5-HT3 Receptor

The role of the 5-HT3 ionotropic receptor in the nociceptive processes of fibromyalgia (FM) patients remains elusive. However, the use of antagonists such as tropisetron has shown promising results in alleviating pain in FM sufferers, whether administered orally or intravenously. Current research suggests that a 10-day oral regimen of 5 mg of tropisetron per day yields positive results. Intravenous administration for 1 to 5 days has been effective in reducing pain, particularly at higher doses. In addition to pain relief, tropisetron has been reported to ameliorate other FM-related symptoms, including sleep disturbances, anxiety, and depression [49,52].

5-HT3 antagonists can mitigate pain by blocking the release of neurotransmitters such as substance P and calcitonin gene-related peptide from nociceptors [53], which are involved in nociception and can trigger peripheral inflammatory responses. Overexpression of 5-HT3 receptors in the CNS regions responsible for pain processing implies that these antagonists could influence the central processing of sensory stimuli, offering a potential therapeutic avenue to manage the symptoms of FM.

### 4.4. Selective Serotonin Reuptake Inhibitors

The role of serotonin (5-HT) in pain neuromodulation underpins the rationale for the use of selective serotonin reuptake inhibitors (SSRIs) in treating fibromyalgia (FM). However, the effectiveness of SSRIs in FM treatment has yielded mixed results in controlled clinical trials (Table 1).

An evaluation of six studies encompassing 343 participants demonstrated that SSRIs were significantly more effective than placebos in achieving at least a 30% reduction in pain [54]. Although some evidence suggests that SSRIs can reduce pain in patients with FM, the overall quality of this evidence remains low. In a more recent meta-analysis, duloxetine, an SSRI, was compared with a placebo in various controlled trials. This analysis indicated that duloxetine outperformed placebo in pain relief, as measured by patient-reported outcomes such as the Fibromyalgia Impact Questionnaire and the Brief Pain Inventory. However, the exact percentage of pain reduction was not specified, and the study highlighted the need for individualized dosing of duloxetine for FM treatment, underscoring the need for customized therapeutic approaches [55]. On the other hand, the dose and duration of medication in the treatment of fibromyalgia with SSRIs can be crucial for observing effective therapeutic outcomes. An appropriate dose ensures that the drug reaches sufficient levels in the body to affect neurotransmitters, such as serotonin, involved in pain modulation. However, the effectiveness of the treatment depends not only on the dose but also on the duration of medication administration. In the early stages of treatment, the drug may not have reached a stable concentration in the body, which could explain the lack of short-term effects, as observed in studies where the medication showed a notable effect at 21 weeks but not at 9 weeks. Over time, the accumulation of the drug can improve serotonin modulation and, consequently, enhance the treatment’s effectiveness in reducing pain.

Despite the theoretical foundation for using SSRIs in FM treatment on the basis of decreased serotonin levels, the precise relationship between 5-HT modulation by these drugs and pain relief remains to be fully clarified [56,57].

-**NCT01288807**: Conducted in eight adult female patients, this open study assessed milnacipran at 200 mg daily over 12 weeks but did not report no analgesic effect.-**NCT00314249**: This large-scale, multicenter, double-blind, placebo-controlled study involving 1025 adult FM patients evaluated milnacipran at 100 mg per day for 12 weeks and reported a positive analgesic effect.-**NCT00115804**: A pilot trial in six patients with juvenile primary fibromyalgia syndrome treated with fluoxetine (10 to 60 mg daily for 12 weeks) reported a decrease in the analgesic effect.-**NCT01237587**: In 184 patients with juvenile primary fibromyalgia syndrome, this quadruple-blind, placebo-controlled study assessed duloxetine (30/60 mg/day for 23 weeks) and found a positive analgesic effect.-**NCT01552057**: A randomized, double-blind, placebo-controlled trial with 393 adult patients with FM that evaluated duloxetine at 60 mg/day for 15 weeks and reported a positive outcome.-**NCT00797797**: This study combined milnacipran (100 mg/day) and pregabalin (300 or 450 mg/day) over 11 weeks in 364 adult FM patients and reported a positive analgesic effect.-**NCT01108731** and **NCT01173055**: These studies, which assessed milnacipran at lower doses (12.5 mg for 9 weeks and 200 mg/day, respectively), reported no analgesic effects in their small cohorts.-**NCT00673452**: A large trial of 530 adult FM patients evaluating duloxetine (60–120 mg/day for 12 weeks), which revealed a positive analgesic effect.-**NCT01077375**: This multicenter study assessed milnacipran (100 to 200 mg/day for 10 weeks) in 120 adult FM patients and reported a positive analgesic effect.-**NCT01829243**: Focusing on older adult FM patients (26 participants), this study assessed milnacipran at various doses (12.5–200 mg/day) and found no analgesic effect.-**NCT00965081**: A quadruple-blind study with 308 adult FM patients evaluated duloxetine at 30 mg/day for 12 weeks and reported a positive effect.-**NCT01038323**: Milnacipran at 12.5 mg/day for 21 weeks was administered to 58 adult FM patients, and a positive analgesic effect was detected.

These trials collectively highlight the variable efficacy of SSRIs in FM treatment, highlighting the importance of personalized approaches to dosage and treatment duration.

## 5. Role of Kynurenine in Pain in Fibromyalgia

Kynurenine, a metabolite of the amino acid tryptophan (Trp), is produced through catabolism initiated by the enzymes indoleamine 2,3-dioxygenase (IDO) and tryptophan 2,3-dioxygenase (TDO), which catalyze the conversion of Trp into N-formyl kynurenine. This process subsequently involves several steps leading to the formation of kynurenine, a process pivotal for numerous biological functions, including immune response regulation, neurotransmission, and cellular equilibrium [21].

Dysfunction in the kynurenine pathway has been associated with various neurological and psychiatric conditions and with the development of chronic pain [58]. The metabolites produced by this pathway, quinolinic acid (QA) and kynurenic acid (KYNA), have different impacts on neuronal functions, inflammation, and oxidative stress, contributing to diseases such as Alzheimer’s disease, Parkinson’s disease, and depression, as well as modulating pain perception, especially in chronic pain scenarios [58].

KYNA is known for its antinociceptive properties through glutamate modulation in the CNS, while QA facilitates pain by acting as an excitotoxin, suggesting that its balance is critical for pain perception and processing [21].

There is a suggested link between kynurenine pathway dysfunction and altered serotonin metabolism, neuroplasticity issues, and neurological disorders of specific relevance to chronic pain due to its interaction with N-methyl-D-aspartate (NMDA) receptors, which are essential for pain signal transduction [21].

Research by Maganin et al. (2022) highlighted that peripheral nerve injury triggers IDO1 upregulation in dendritic cells (DCs), leading to increased SC kynurenine. This kynurenine is then converted by astrocytic kynurenine 3-monooxygenase (KMO) into the pronociceptive metabolite 3-hydroxykynurenine (3-HK). Furthermore, DCs expressing 3-hydroxyanthranilic acid are involved in increasing spinal cord QA, promoting pronociceptive actions by enhancing glutamatergic transmission and sustaining pain hypersensitivity [59].

While KYNA is recognized for its neuroprotective and pain-modulating effects, QA is considered neurotoxic, contributing to neuroinflammation and excitotoxicity, which are pivotal in pain signaling. In addition, cerebral IDO activity has been associated with the comorbidity of pain and depression. The role of kynurenine in major depression via glutamatergic transmission modulation suggests its potential involvement in pain processing. Positive regulation of neuronal KMO has been shown to induce depression-like behavior in neuropathic pain models in mice, indicating its potential influence on pain-related behavioral changes (Figure 1) [60,61].

## 6. Discussion

Given the essential role of tryptophan (Trp) as a dietary amino acid and precursor in several key metabolic pathways, including those leading to serotonin (5-HT) and niacin (vitamin B3), as well as its involvement in the kynurenine pathway crucial for immune modulation and synthesis of nicotinamide adenine dinucleotide (NAD+), as an enzyme vital for cell energy production, Trp stands at the intersection of numerous studies related to cognitive function and behavior [62].

Trp is an integral part of the neuromodulation and catabolism of proteins during trauma, infection, and cancer. Studies suggest that Trp levels may be decreased in serum from patients with fibromyalgia (FM), potentially impairing immunomodulatory functions and leading to symptoms such as severe fatigue, sleep disturbances, cognitive problems, and pain [62,63].

A landmark study by King (1980) [64] on the analgesic properties of Trp in postsurgical patients revealed that Trp administration reduced recurring pain and sensory deficits, highlighting Trp’s role in neurotransmitter modulation and its potential to address pain and sensory issues through the serotonin pathway.

Recent findings indicate reduced levels of Trp and kynurenine in FM patients, suggesting a link between these metabolites and the severity of FM symptoms [36]. The rapid release of cortisol in response to emotional stress, which activates the Trp–kynurenine pathway and reduces Trp availability for 5-HT synthesis, underscores the complex interactions affecting Trp metabolism under stress and disease conditions [65].

These findings collectively suggest that altered Trp metabolism in FM affects the balance between 5-HT and kynurenine and potentially impacts the central and peripheral nervous systems. Such alterations could contribute to the pathophysiology of pain in FM, emphasizing the need for further research to fully understand these mechanisms and their implications for treatment (Figure 2).

Research by Russell, Michalek, et al. and Russell, Vaeroy, et al. in 1992 [23] elucidated potential aberrations in the descending pain regulatory pathways as contributors to the painful symptoms experienced by patients with fibromyalgia (FM). These pathways, which are crucial for the inhibition of the transmission of sensory stimuli from the brainstem to the dorsal root, leverage neurotransmitters linked to pain and mood regulation, including serotonin (5-HT).

FM patients exhibit heightened pain sensitivity correlated with increased sensory input, requiring approximately 50% less stimulus intensity to elicit pain than healthy individuals [8,66]. Dysfunctions in the autonomic nervous system, which are commonly observed in FM patients, might exacerbate sensitivity and other clinical manifestations by disrupting the physiological responses essential for stress management and pain inhibition, which are mediated by growth hormone and insulin growth factor 1 (IGF-1). Although 5-HTT has been implicated in heightened pain sensitivity among FM patients [8], research by Gursoy (2002) [67] suggested that the influence of 5-HT on pain perception is not mediated by transporter-regulating genes but by those regulating metabolic enzymes and receptors such as 5-HT2A and 5-HT3, with clinical trials affirming the effectiveness of 5-HT3 antagonists in FM treatment [68].

The role of 5-HT in pain regulation varies according to the receptors engaged; for example, the 5-HT1A interaction yields antinociceptive effects, whereas 5-HT2A and 5-HT3 agonists contribute to pronociception [69]. Given this diversity, the ability of 5-HTT to regulate 5-HT levels in the synaptic cleft may not specifically inhibit pain perception. This complexity renders SSRIs alone an insufficient and reductionist treatment approach for FM, underscored by mixed clinical outcomes and the potential of inflammation as a common denominator in chronic pain and depression. Inflammation triggers the kynurenine pathway, a crucial metabolic process of Trp, leading to the production of neuroactive metabolites [70].

The kynurenine pathway, which is regulated primarily by proinflammatory cytokines such as TNF-α and IFN-γ, progresses to KYNA synthesis by astroglia and QA by microglia. The balance between neuroprotective KYNA and neurotoxic QA is pivotal, with recent studies linking diminished KYNA levels and elevated QA to chronic pain and depression pathogenesis, suggesting a potential weak effect of 5-HT on these conditions in FM [70,71,72].

Therefore, we propose targeting Trp metabolism as a treatment strategy for FM, focusing on inhibiting neurotoxic QA metabolism without excluding SSRIs. Enhancing KYNA could provide neuroprotection and analgesic benefits while indirectly increasing 5-HT levels to inhibit nociceptive signals (Figure 3). This comprehensive approach may pave the way for innovative FM treatments, fostering research into new pharmaceuticals and significantly enhancing the quality of life for FM patients by effectively managing their pain and depression.

## 7. Conclusions

Recent studies have illuminated the altered metabolism of tryptophan (Trp) in fibromyalgia (FM) patients, revealing notably low levels of serotonin (5-HT), attributable to the predominant metabolism of Trp through the kynurenine pathway. This pathway, which is responsible for the synthesis of metabolites implicated in FM, diverts up to 95% of Trp, thus impacting 5-HT synthesis.

These findings challenge the efficacy of treatments exclusively utilizing selective serotonin reuptake inhibitors (SSRIs) for FM pain management. Thus, targeting 5-HT alone is insufficient for effective pain inhibition, and the metabolites of the kynurenine pathway should not be viewed solely as nociceptive facilitators.

## 8. Perspectives

This review underscores the complex interplay among neuronal, immune, and metabolic pathways in FM, which contributes to the manifestation of pain and depression. We propose a novel treatment strategy aimed at reducing symptoms by inhibiting quinolinic acid (QA) synthesis alongside SSRI administration. This approach aims to increase the availability of both 5-HT and kynurenic acid (KYNA), promoting neuroprotection and analgesia. This multifaceted treatment modality acknowledges the intricacies of Trp metabolism in FM and represents a step forward in the development of more comprehensive and effective therapeutic interventions.

## Figures and Tables

**Figure 1 pharmaceuticals-17-01205-f001:**
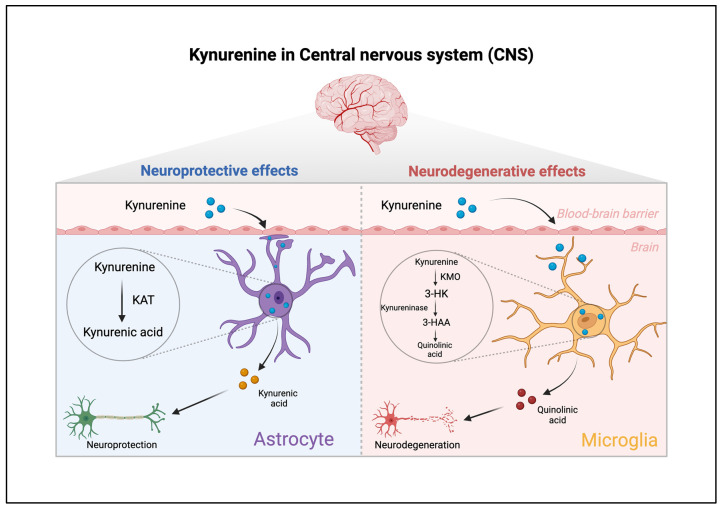
Dual roles of metabolites within the kynurenine pathway, focusing on quinolinic acid and its contrast to quinuric acid in the central nervous system. Quinuric acid, which is synthesized by astrocytes from quinolinic acid, is the result of the enzymatic action of kynurenine aminotransferase (kat). This conversion process highlights the neuroprotective properties of quinuric acid. Conversely, quinolinic acid, when processed by microglia, follows a different metabolic route involving the kynurenine monooxygenase (kmo) signaling cascade. Kmo facilitates the conversion of quinolinic acid to 3-hydroxykynurenine (3-hk). The pathway continues with the transformation of 3-hk into 3-hydroxyanthranilic acid (3-haa) by the enzyme quinureninase, eventually leading back to the formation of quinolinic acid, which is known for its neurodegenerative effects. This depiction underscores the complex interplay between neuroprotective and neurodegenerative mechanisms mediated by metabolites of the kynurenine pathway.

**Figure 2 pharmaceuticals-17-01205-f002:**
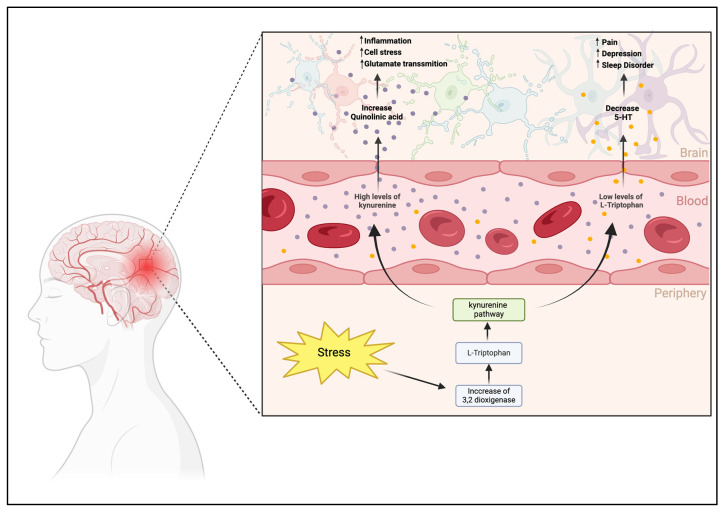
The impact of L-tryptophan on fibromyalgia pathology underscores its crucial role through its metabolic products, serotonin, and kynurenine. Stress conditions prompt an increase in 3,2-dioxygenase activity, which favors the kynurenine pathway over serotonin synthesis by depleting the L-tryptophan levels in the bloodstream. Consequently, in the central nervous system (CNS), decreased availability of L-tryptophan leads to decreased serotonin levels, which can cause symptoms such as depression, pain, and sleep disturbances characteristic of fibromyalgia. Concurrently, an increase in kynurenine levels results in elevated quinolinic acid within the CNS, hinting at a neurodegenerative process propelled by inflammation, cellular stress, and increased glutamatergic activity. These interconnected pathways could contribute to the chronic pain and central sensitization observed in fibromyalgia patients, indicating that modulation of L-tryptophan and kynurenine levels is a promising direction for therapeutic intervention in the treatment of fibromyalgia symptoms.

**Figure 3 pharmaceuticals-17-01205-f003:**
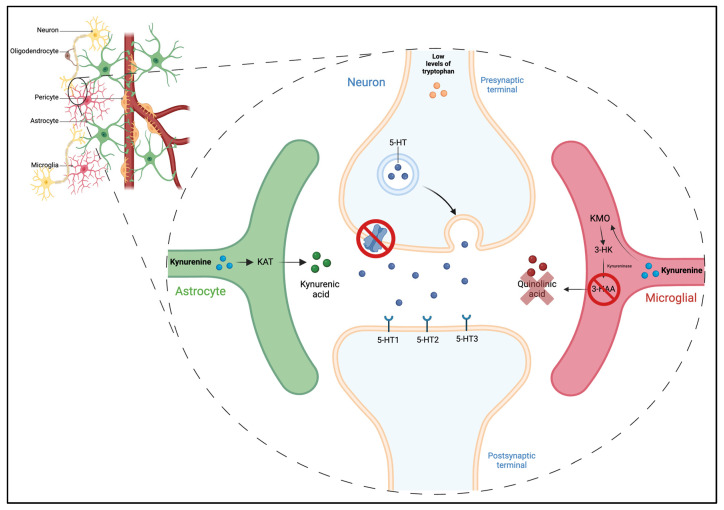
Proposed intervention to mitigate pain in fibromyalgia (FM) patients, which focuses on modulating the serotonin–kynurenine pathway. This strategy involves two principal components: the administration of serotonin reuptake inhibitors (SRIs) to increase the serotonin levels in the synaptic cleft and the inhibition of 3-hydroxyanthranilic acid (3-HAA) production in microglia, thereby curtailing the synthesis of quinolinic acid. These targeted actions are designed to exert beneficial effects on pain alleviation in FM sufferers. Enhancing serotonin availability through SRIs is anticipated to improve the functionality of nociceptive pathways, potentially diminishing pain perception. Additionally, serotonin elevation may ameliorate depression symptoms, which are frequently observed in FM patients, thereby tackling both the neurochemical and the psychological dimensions of the condition. Simultaneously, the proposed intervention seeks to reduce quinolinic acid—a metabolite linked to neurodegenerative effects—and increase quinuric acid, which is known for its neuroprotective properties within the central nervous system. By decreasing the presence of quinolinic acid, it is hypothesized that there would be a concomitant reduction in the pain experienced by FM patients. This dual-focused approach aims not only to alleviate the immediate symptoms of pain but also to address the underlying neurochemical imbalances associated with FM, providing a holistic treatment strategy that encompasses both the physical and the psychological aspects of the disease.

**Table 1 pharmaceuticals-17-01205-t001:** Comprehensive overview of clinical trials investigating the efficacy of serotonin reuptake inhibitors (SRIs) as analgesic agents in patients with fibromyalgia (FM). Trials vary widely in design, demographics of the patients, dosage of medications and outcomes, providing a broad perspective on the analgesic effect of SRIs in FM treatment.

Registry Number	Patients	Study Design	Medication	Dose	Analgesic Effect
NCT01288807	8 adult female patients with fibromyalgia	Single group assignment, open study	Milnacipran	200 mg/day for 12 weeks	NO
NCT00314249	1025 adult patients with fibromyalgia	Randomized, multicenter, double-blind, placebo controlled	Milnacipran	100 mg/day for 12 weeks	YES
NCT01173055	22 adult female patients with fibromyalgia	Randomized, double-blind, two-way placebo controlled	Milnacipran	200 mg/day for 6 weeks	NO
NCT01108731	37 adult fibromyalgia patients	Quadruple-blind, randomized, placebo controlled	Milnacipran	12.5 mg/day for 9 weeks	NO
NCT01077375	120 adult patients with fibromyalgia	Multicenter, randomized, double-blind, placebo controlled	Milnacipran	100 to 200 mg/day for 10 weeks	YES
NCT01829243	26 older adult patients with fibromyalgia	Randomized, double-blind, two-way placebo controlled	Milnacipran	12.5 mg 200 mg/day for 13 weeks	NO
NCT01038323	58 adult fibromyalgia patients	Triple-blind, randomized, 3-way placebo controlled	Milnacipran	12.5 mg/day for 21 weeks	YES
NCT00797797	364 adult fibromyalgia patients	Multicenter, randomized, open-label, placebo controlled study	Milnacipran and pregabalin	Milnacipran 100 mg/day and pregabalin 300 or 450 mg/day for 11 weeks	YES
NCT00673452	530 adult fibromyalgia patients	Quadruple-blind, randomized, two-way placebo controlled	Duloxetine	60–120 mg/day for 12 weeks	YES
NCT01237587	184 patients with juvenile primary fibromyalgia syndrome	Multicenter, quadruple-blind, randomized, placebo-controlled, 2-way crossover	Duloxetine	30/60 mg/day for 23 weeks	YES
NCT00965081	308 adult fibromyalgia patients	Quadruple-blind, randomized, two-way placebo controlled	Duloxetine	30 mg/day for 12 weeks	YES
NCT01552057	393 adult fibromyalgia patients	Randomized, double-blind, placebo controlled	Duloxetine	60 mg/day for 15 weeks	YES
NCT00115804	6 patients with juvenile primary fibromyalgia syndrome	Pilot trial, single group assignment, open study	Fluoxetine	10 to 60 mg/day for 12 weeks	LOWER

## Data Availability

No new data were used for the research described in this paper.

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
