# Peer review of "Association of the Serotonin and Kynurenine Pathways as Possible Therapeutic Targets to Modulate Pain in Patients with Fibromyalgia"

_pharmaceuticals, 2024, doi:10.3390/ph17091205_

Round 1

Reviewer 1 Report

Comments and Suggestions for Authors

After reading the article Association of the serotonin and kynurenine pathways as possible therapeutic targets to modulate pain in fibromyalgia, where the authors want to highlight new therapeutic alternatives for the management of fibromyalgia related to serotonin and the serotonin pathway. kynurenine, I found the information captured in this review to be very interesting.

However, I suggest that the authors attach information about the search for the articles, what their criteria were, etc.

Is there information at a preclinical level?

Be more specific in the conclusion.

Attach an perspectives section.

In the references section, correct the name of the journals in italics and the year in bold.

Comments on the Quality of English Language

Minor editing of English language required

Author Response

Thank you very much for taking the time to review this manuscript. Below you will find detailed responses and any relevant revisions or corrections underlined in yellow in the forwarded file.

COMMENT 1: I suggest that authors attach information about the search for the articles, what their criteria were, etc.

RESPONSE 1: Information attached at the end of the introduction.

COMMENT 2: Is there information at a preclinical level?

RESPONSE 2: Yes, the text presents information at a preclinical level, especially in the section related to the pathophysiology and treatment of pain in fibromyalgia. For example:

  1. Research on pain and depression: It mentions that "Postmortem analyses have revealed reductions in neuronal bodies within the prefrontal cortex, hippocampus, and thalamus, indicative of diminished synaptic activity" and refers to studies suggesting neuroplastic alterations in animal models or postmortem analysis, which is a preclinical level.
  2. Studies with mirtazapine: In the section "4. Role of serotonin receptors in pain in patients with fibromyalgia", it mentions that "Research involving mirtazapine has demonstrated its potential in reversing chronic pain and improving the analgesic effects of morphine in FM-like mouse models". This is a clear example of studies preclinical studies in animal models investigating the effectiveness of treatments for fibromyalgia.

These references indicate that preclinical studies have been conducted to explore both the pathophysiology of fibromyalgia and possible treatments, which is crucial to better understand the disease before proceeding to clinical trials in humans.

COMMENT 3: Be more specific in the conclusion.

RESPONSE 3: The conclusion was integrated, and the perspectives section was separated.

COMMENT 4: Attach a perspectives section.

RESPONSE 4: The perspectives section that was within the conclusion was placed.

COMMENT 5: In the references section, correct the name of the journals in italics and the year in bold.

RESPONSE 5: The observation is considered, and the text is corrected.

Finally , we would like to express our sincere gratitude to the reviewer of this article for their valuable insights and suggestions. Their meticulous review and constructive feedback have significantly contributed to the improvement of our manuscript. We deeply appreciate the time and effort invested in reviewing our work, which has greatly enhanced the quality and scientific rigor of the paper. Their comments have been instrumental in refining our analyses and improving the clarity of our conclusions, thereby strengthening the impact of our research.

Reviewer 2 Report

Comments and Suggestions for Authors

This review proposes a novel therapeutic approach for FM pain management, focusing on inhibiting QA synthesis while co-administering selective serotonin reuptake inhibitors (SSRIs) to potentially elevate KYNA levels. This dual approach aims to dampen pain perception and improve patient outcomes. Although this review presents relevant and promising insights, several adjustments are necessary to enhance its comprehensiveness and accuracy.

Major Suggestions:

Point 1: Although the introduction provides a concise summary of fibromyalgia, it does not adequately address the relationship between this condition and the pathway of interest in this review. Additionally, some sentences containing important information about epidemiological data and treatments lack references. For example, the following statement is unreferenced: "The economic impact is substantial, with medical and home care expenses reaching approximately $500 million annually. In addition, the primary treatment strategy, which often includes opioid administration, has been associated with a significant decline in quality of life over time." The author presents data limited to the United States; it is important to consider including updated information on the global context of fibromyalgia.

Point 2: Are opioids still a primary strategy for the treatment of fibromyalgia? Current studies indicate that the use of opioids in the management of fibromyalgia is not recommended because, in addition to causing dependence and tolerance, it can exacerbate pain hypersensitivity (doi: 10.1097/j.pain.0000000000002878; 10.1097/RHU.00000000000001273). The American College of Rheumatology does not recommend this treatment. I suggest reviewing this information.

Point 3: Regarding the pathophysiology of fibromyalgia, it is necessary to briefly discuss the main mechanisms currently considered in the pathogenesis of the disease, highlighting the dysfunction of the monoamine pathway and linking it to the serotonin deficit.

Point 4: In the section "Role of Serotonin Receptors in Pain in Fibromyalgia," I suggest adding a summary of the role of serotonin in the descending inhibitory pathway of pain. This would better explain the importance of this neurotransmitter in this pathway and the consequences of its dysfunction.

Point 5: On the topic of "Selective Serotonin Reuptake Inhibitors," the duration of treatment in each study, as well as the age and sex of the patients, may have been decisive factors in the drug's effectiveness. For example, the table shows that daily treatment with Milnacipran at a dose of 12.5 mg/day had an effect over 21 weeks, but not over 9 weeks. To what can this difference be attributed? Adding a paragraph directly relating these specifics and presenting studies that support this observation could help justify the search for new therapies and the study of different pathways.

Point 6: I suggest reorganizing the table, as the information is currently confusing. Some studies do not include treatment time, and others do not specify the sex of the patients. It is worth separating the information in the "Medication/Dose" column by creating an additional column for treatment time. Additionally, grouping information with the same drug and dose in sequence would facilitate comparison.

Point 7: What is the signaling pathway of the 5-HT1, 2A, and 3 receptors? Are they ionotropic or metabotropic receptors? Adding this information to the manuscript would be relevant.

Comments on the Quality of English Language

Minor editing of English language required.

Author Response

Thank you very much for taking the time to review this manuscript. Below you will find detailed responses and any relevant revisions or corrections underlined in yellow in the forwarded file.

COMMENT 1: Although the introduction provides a concise summary of fibromyalgia, it does not adequately address the relationship between this condition and the pathway of interest in this review. Additionally, some sentences containing important information about epidemiological data and treatments lack references. For example, the following statement is unreferenced: "The economic impact is substantial, with medical and home care expenses reaching approximately $500 million annually. In addition, the primary treatment strategy, which often includes opioid administration, has been associated with a significant decline in quality of life over time." The author presents data limited to the United States; it is important to consider including updated information on the global context of fibromyalgia.

RESPONSE 1: We have corrected the missing reference and added a paragraph providing a global perspective on fibromyalgia, considering it important for the introduction. In addition, the relationship between the proposed pathway and its connection with fibromyalgia is detailed in later sections. For this reason, and to avoid redundancies and confusion in the reader, we believe it is appropriate not to include an additional paragraph in the introduction.

COMMENT 2: Are opioids still a primary strategy for the treatment of fibromyalgia? Current studies indicate that the use of opioids in the management of fibromyalgia is not recommended because, in addition to causing dependence and tolerance, it can exacerbate pain hypersensitivity (doi: 10.1097/j.pain.0000000000002878; 10.1097/RHU.00000000000001273). The American College of Rheumatology does not recommend this treatment. I suggest reviewing this information.

RESPONSE 2: Thank you for pointing this out. We confirm that opioids are indeed not recommended for the treatment of fibromyalgia. Recent studies indicate that opioid use may cause dependence, tolerance, and exacerbation of pain hypersensitivity.

We have changed the wording of the following paragraph in the introduction:

“Furthermore, the primary treatment strategy, which often includes opioid administration, has been associated with a significant decline in quality of life over time.”

It has now been changed to: “The primary treatment for fibromyalgia to inhibit pain involves a combination of pharmacological approaches, physical therapies, psychological support, and self-management strategies. This multimodal treatment plan is designed to address the diverse symptoms experienced by patients with fibromyalgia, ultimately improving their quality of life. These observations highlight the urgent need for improved diagnostic and therapeutic approaches for FM and related disorders.”

COMMENT 3: Regarding the pathophysiology of fibromyalgia, it is necessary to briefly discuss the main mechanisms currently considered in the pathogenesis of the disease, highlighting the dysfunction of the monoamine pathway and linking it to the serotonin deficit.

RESPONSE 3: The comment was considered, and the following paragraph was added in the Physiopathology of fibromyalgia section (underlined in yellow):

FM is closely associated with dysfunction in monoamine pathways, particularly concerning serotonin. Alterations in neurotransmitter systems, such as serotonin (5-HT) and norepinephrine (NE), are implicated in mood and pain disorders. Genetic polymorphisms, such as those in the serotonin transporter (5-HTTLPR), affect serotonin signaling, increasing susceptibility to conditions like fibromyalgia and depression. Neuroinflammation and dysregulation of the hypothalamic-pituitary-adrenal (HPA) axis also contribute to exaggerated pain responses and mood disorders, linking inflammation to monoamine dysfunction. Additionally, functional imaging studies reveal that certain serotonin-related genetic variants are associated with increased activation in brain regions responsible for pain perception. These mechanisms highlight the importance of the balance between neurotransmitters such as serotonin, norepinephrine, and dopamine, and their impact on fibromyalgia [8].

COMMENT 4: In the section "Role of Serotonin Receptors in Pain in Fibromyalgia," I suggest adding a summary of the role of serotonin in the descending inhibitory pathway of pain. This would better explain the importance of this neurotransmitter in this pathway and the consequences of its dysfunction.

RESPONSE 4: The comment was considered, and the following paragraph was added in the Physiopathology of fibromyalgia section (underlined in yellow):

The descending pain pathway is a complex neural system crucial for pain modulation, as it transmits signals from the brain to the spinal cord and influences pain perception. This pathway involves nociceptive transmission, where painful stimuli are conveyed from peripheral receptors to the brain for conscious perception. A key component of this modulation is serotonin, a neurotransmitter that inhibits pain signals in the spinal cord by reducing the release of substance P and regulating nociceptive responses. In conditions such as fibromyalgia, lower levels of serotonin are associated with increased pain sensitivity, underscoring the role of serotonin in pain modulation [35].

COMMENT 5: On the topic of "Selective Serotonin Reuptake Inhibitors," the duration of treatment in each study, as well as the age and sex of the patients, may have been decisive factors in the drug's effectiveness. For example, the table shows that daily treatment with Milnacipran at a dose of 12.5 mg/day had an effect over 21 weeks, but not over 9 weeks. To what can this difference be attributed? Adding a paragraph directly relating these specifics and presenting studies that support this observation could help justify the search for new therapies and the study of different pathways.

RESPONSE 5: Thank you for your comment. We have reviewed the duration of treatment as well as the age and sex of patients in the included studies, recognizing that these factors may significantly influence the efficacy of selective serotonin reuptake inhibitors (SSRIs) in the treatment of fibromyalgia (FM). For example, as you mention, daily treatment with Milnacipran at a dose of 12.5 mg/day showed a noticeable effect at 21 weeks, but not at 9 weeks. This difference could be attributed to the gradual build-up of the drug in the system, allowing for more effective modulation of serotonin levels over time. In addition, factors such as variability in patient response and adaptation of the central nervous system to the effects of the drug over time could also play a role.

The following paragraph was added to the text to clarify these details:

On the other hand, the dose and duration of medication in the treatment of fibromyalgia with SSRIs can be crucial for observing effective therapeutic outcomes. An appropriate dose ensures that the drug reaches sufficient levels in the body to affect neurotransmitters, such as serotonin, involved in pain modulation. However, the effectiveness of the treatment depends not only on the dose but also on the duration of medication administration. In the early stages of treatment, the drug may not have reached a stable concentration in the body, which could explain the lack of short-term effects, as observed in studies where the medication showed a notable effect at 21 weeks but not at 9 weeks. Over time, the accumulation of the drug can improve serotonin modulation and, consequently, enhance the treatment's effectiveness in reducing pain.

COMMENT 6: I suggest reorganizing the table, as the information is currently confusing. Some studies do not include treatment time, and others do not specify the sex of the patients. It is worth separating the information in the "Medication/Dose" column by creating an additional column for treatment time. Additionally, grouping information with the same drug and dose in sequence would facilitate comparison.

RESPONSE 6: The comment is considered, and the columns for dose and medication are separated, and the rows are unified by medication.

COMMENT 7: What is the signaling pathway of the 5-HT1, 2A, and 3 receptors? Are they ionotropic or metabotropic receptors? Adding this information to the manuscript would be relevant.

RESPONSE 7: Thanks for the comment, 5-HT1 and 5-HT2A receptors are metabotropic and coupled to G proteins. The 5-HT1 receptor is associated with the Gi/Go protein, leading to inhibition of adenylate cyclase and decreased cAMP levels, modulating functions such as mood regulation and pain perception. The 5-HT2A receptor, on the other hand, couples to the Gq/11 protein, activating phospholipase C and subsequently increasing intracellular calcium concentration and activating protein kinase C, processes that influence neurotransmission and smooth muscle contraction. The 5-HT3 receptor is ionotropic and functions as a ligand-gated ion channel, allowing the flow of cations and leading to rapid depolarization of neuronal membranes, resulting in rapid neuronal responses, including pain transmission. This information will be included briefly within the document.

Finally, we would like to express our sincere gratitude to the reviewer of this article for their valuable insights and suggestions. Their meticulous review and constructive feedback have significantly contributed to the improvement of our manuscript. We deeply appreciate the time and effort invested in reviewing our work, which has greatly enhanced the quality and scientific rigor of the paper. Their comments have been instrumental in refining our analyses and improving the clarity of our conclusions, thereby strengthening the impact of our research

Reviewer 3 Report

Comments and Suggestions for Authors

The manuscript aims to review the relationship between tryptophan metabolism and fibromyalgia (which does not totally agree with the title). Yet, after reading the review, the only message obtained is: yes, the paper claims that there may be a relationship based on literature; whereas what is this relationship,  what is the evidences, can you trust them? You really need to go to the 71 citations (by mistake the 72nd citation is the authors’ disclaimer). Here is an example:

Simply writing as “Evidence suggests…” (e.g., line 86) does not help and it is hard for the readers to check all citations to figure out what are the evidence (that "suggest") and whether they are convincing or not.

Comments on the Quality of English Language

The language is easy to understand.

Author Response

Thank you very much for taking the time to review this manuscript.

COMMENT 1: The manuscript aims to review the relationship between tryptophan metabolism and fibromyalgia (which does not totally agree with the title). Yet, after reading the review, the only message obtained is: yes, the paper claims that there may be a relationship based on literature; whereas what is this relationship, what is the evidences, can you trust them? You really need to go to the 71 citations (by mistake the 72nd citation is the authors’ disclaimer). Here is an example:

Simply writing as “Evidence suggests…” (e.g., line 86) does not help and it is hard for the readers to check all citations to figure out what are the evidence (that "suggest") and whether they are convincing or not.

RESPONSE 1: Thank you for the comment, but we consider that the manuscript in its current form meets the objective of reviewing the relationship between tryptophan metabolism and fibromyalgia. The text already includes a detailed analysis of the relevant literature and is designed to provide an overview that allows readers to understand the complexity of this relationship.

Although we recognize that some statements could be more detailed, the current structure of the paper allows the interested reader to delve deeper into specific references as needed. This also avoids overloading the manuscript with excessive information, maintaining a balance between accessibility and depth of content.

Finally, we would like to express our sincere gratitude to the reviewer of this article for their valuable insights and suggestions. Their meticulous review and constructive feedback have significantly contributed to the improvement of our manuscript. We deeply appreciate the time and effort invested in reviewing our work, which has greatly enhanced the quality and scientific rigor of the paper. Their comments have been instrumental in refining our analyses and improving the clarity of our conclusions, thereby strengthening the impact of our research

Round 2

Reviewer 3 Report

Comments and Suggestions for Authors

The manuscript is improved